# Brief Hospital Supervision of Exercise and Diet During Adjuvant Breast Cancer Therapy Is Not Enough to Relieve Fatigue: A Multicenter Randomized Controlled Trial

**DOI:** 10.3390/nu12103081

**Published:** 2020-10-09

**Authors:** William Jacot, Antoine Arnaud, Marta Jarlier, Claudia Lefeuvre-Plesse, Philippe Dalivoust, Pierre Senesse, Ahmed Azzedine, Olivier Tredan, Sophie Sadot-Lebouvier, Sébastien Mas, Marion Carayol, Jean-Pierre Bleuse, Sophie Gourgou, Chloé Janiszewski, Silene Launay, Véronique D’Hondt, Géraldine Lauridant, Julien Grenier, Gilles Romieu, Gregory Ninot, Laurence Vanlemmens

**Affiliations:** 1Val d’Aurelle Montpellier Cancer Institute (ICM), 208 Avenue des Apothicaires, Parc Euromédecine, CEDEX 5, 34298 Montpellier, France; marta.jarlier@icm.unicancer.fr (M.J.); pierre.senesse@icm.unicancer.fr (P.S.); sebastien.mas@protonmail.com (S.M.); jean-pierre.bleuse@icm.unicancer.fr (J.-P.B.); sophie.gourgou@icm.unicancer.fr (S.G.); chloe.janiszewski@icm.unicancer.fr (C.J.); silene.launay@icm.unicancer.fr (S.L.); veronique.dhondt@icm.unicancer.fr (V.D.); gilles.romieu@icm.unicancer.fr (G.R.); gregory.ninot@umontpellier.fr (G.N.); 2Faculty of Medicine, University of Montpellier, Rue du Pr. Henri Serre, 34000 Montpellier, France; 3Sainte-Catherine Institute, 1750 Chemin Lavarin, 84000 Avignon, France; a.arnaud@isc84.org (A.A.); j.grenier@isc84.org (J.G.); 4Eugène Marquis Center, Rue de la Bataille Flandres-Dunkerque, CS 44229, 35042 Rennes, France; c.lefeuvre@rennes.unicancer.fr; 5Ambroise Paré Hospital, 1 Rue de l’Eylau, 13006 Marseille, France; philippedalivoust@gmail.com; 6Montélimar Hospital, Quartier Beausseret, BP 249-26, 26216 Montélimar, France; ahmed.azzedine@gh-portesdeprovence.fr; 7Léon Bérard Center, 28 Rue Laennec, 69008 Lyon, France; olivier.tredan@lyon.unicancer.fr; 8René Gauducheau Center, Boulevard Jacques Monod, 44805 Saint-Herblain, France; sophie.sadot-lebouvier@ico.unicancer.fr; 9IAPS Laboratory “Impact of Physical Activity on Health”, University of Toulon, Avenue de l’Université, 83957 La Garde, France; marion.carayol@univ-tln.fr; 10Oscar Lambret Center, 3 Rue Frédéric Combemale, 59000 Lille, France; g-lauridant@o-lambret.fr (G.L.); l-vanlemmens@o-lambret.fr (L.V.)

**Keywords:** breast cancer, exercise, diet, education, fatigue, weight, quality of life

## Abstract

Supervised exercise dietary programs are recommended to relieve cancer-related fatigue and weight increase induced by adjuvant treatment of early breast cancer (EBC). As this recommendation lacks a high level of evidence, we designed a multicenter randomized trial to evaluate the impact of an Adapted Physical Activity Diet (APAD) education program on fatigue. We randomized 360 women with EBC who were receiving adjuvant chemotherapy and radiotherapy to APAD or usual care at eight French cancer institutions. Data were collected at baseline, end of chemotherapy, end of radiotherapy, and 6 months post-treatment. The primary endpoint was the general cancer-related fatigue score using the MFI-20 questionnaire. Fatigue correlated with the level of precariousness, but we found no significant difference between the two groups in terms of general fatigue (*p* = 0.274). The APAD arm has a smaller proportion of patients with confirmed depression at the end of follow-up (*p* = 0.052). A transient modification in physical activity levels and dietary intake was reported in the experimental arm. However, a mixed hospital- and home-based APAD education program is not enough to improve fatigue caused by adjuvant treatment of EBC. Cancer care centers should consider integrating more proactive diet–exercise supportive care in this population, focusing on precarious patients.

## 1. Introduction

Cancer-related fatigue is the most distressing and common symptom reported by patients undergoing adjuvant therapy for breast cancer (BC) [1,2,3,4]. Fatigue provokes an increase in sedentary behaviors, modification of dietary intake, metabolic changes, fat mass increase, depression, and anxiety [5], and can alter cancer prognosis and treatment [6,7,8]. Exercise and nutrition programs are recommended by experts and medical societies to relieve cancer-related fatigue during active treatment and to prevent an increase in weight [9,10]. Exercise for cancer patients must include both moderate-intensity aerobic exercise and muscle-strengthening exercises [9], and be regular, frequent (at least 2 h per week), and progressive. Nutrition must aim to maintain a healthy weight, promote eating more plant-based foods, limit red and processed meat, energy-dense foods, salt, sugary drinks, and alcohol, and not rely on dietary supplements [11]. This is a particularly relevant effect in the clinical context of BC, as body mass index (BMI) before and after BC diagnosis, and weight gain after diagnosis were associated with increased mortality in recent meta-analyses [7,8]. Nutritional consultations help manage nutritional disorders that worsen fatigue, such as anemia, diarrhea, nausea, and vomiting [10,12].

The combination of exercise and dietary support has been reported to induce significant weight loss in survivors of early BC (EBC) after adjuvant chemotherapy/radiotherapy [13,14,15]. Four randomized controlled trials (RCTs) have assessed the combination of exercise and diet in EBC patients undergoing adjuvant chemotherapy and/or radiotherapy, but they were designed as pilot trials with less than 30 patients in each randomization group [16,17,18,19]. Therefore, the benefits of an exercise-diet intervention during adjuvant chemotherapy and radiotherapy need to be evaluated in a well-powered and multicenter RCT. We previously reported a monocentric RCT evaluating an adapted physical activity and diet (APAD) intervention during adjuvant treatment of EBC [18,20] and found a beneficial effect on patient-reported outcomes (PROs). However, these results need to be validated in larger, multicenter cohorts in order to evaluate the impact of the heterogeneity of organizations and sociocultural parameters on the results of the intervention.

As social deprivation has been reported to significantly impact cancer risk and program efficacy, this variable needs to be addressed in order to identify the impact of social inequalities on the performance of a given intervention. For example, vulnerable individuals identified using the validated Evaluation of Deprivation and Health Inequalities in Public Health Centers (EPICES) index are more likely already to have cancer, a higher mean BMI, greater prevalence of current smoking, lower adherence to screening programs, and greater standardized mortality ratio compared to non-vulnerable individuals [21,22,23]. Thus, this social grading variable is expected to impact the adherence of patients to supportive care programs and ultimately induce differences in treatment-induced fatigue in this population.

We designed the present multicenter RCT to assess the effect of a combined exercise and diet intervention delivered during a six-cycle adjuvant chemotherapy regimen followed by radiotherapy on cancer-related fatigue in EBC patients. We hypothesized that the combined supervised program would yield beneficial effects compared to usual care on cancer-related fatigue as a primary outcome, especially in patients considered to be more vulnerable in terms of social deprivation. Secondary outcomes include BMI, nutritional parameters, physical abilities, anxiety-depressive symptoms, and health-related quality of life.

## 2. Materials and Methods

### 2.1. Design

The present study was a two-arm, multicenter, randomized, controlled, prospective trial. The APAD2 trial was designed and implemented to evaluate the impact of an exercise and nutrition-based supportive care intervention during 6 months of chemotherapy and radiotherapy on fatigue, evaluated using the MFI-20 questionnaire in EBC patients treated in eight French cancer centers. The APAD intervention was compared to usual care without specific exercise and/or nutrition care (control arm). The primary hypothesis was the possibility of obtaining a 4-point reduction in the mean score on the General fatigue subscale in the intervention group with respect to the control group.

### 2.2. Participants

Eligible participants were women over 18 years of age with histologically proven and newly (<6 months) affected by EBC accessible to initial surgery, without consideration of their baseline physical activity level or dietary intake. Patients were enrolled after undergoing curative surgery. All patients were to receive six cycles of adjuvant chemotherapy (three cycles of epirubicin/cyclophosphamide/5-fluorouracil every 3 weeks (FEC100 protocol), followed by three taxane-based cycles, either docetaxel every 3 weeks [24] or paclitaxel weekly for 9 weeks), followed by 6 weeks of radiotherapy. Patients affected by HER2-positive tumors also received adjuvant trastuzumab for a total of 52 weeks, starting at the initiation of taxane chemotherapy. Exclusion criteria were metastatic disease, any other primary tumor, medical contra-indications to moderate-intensity physical activity, inability to attend intervention sessions or assessments, and a difficulty or disability preventing the patient from correctly understanding the trial information or requirements. The study was approved by the local ethics committee (NCT04109326) and conducted in accordance with the Declaration of Helsinki and principles of good clinical practice.

Before chemotherapy, potential participants were identified by the hospital medical oncologists. All participants were informed of the goal of the study and the potential benefits of diet and exercise on fatigue during adjuvant therapy. The patients who provided written informed consent and completed baseline assessments were randomly assigned (1:1 ratio) to the APAD experimental arm or control arm, stratified by center and precariousness level as assessed by the EPICES score [22] using the minimization method. Randomization was performed at the Montpellier ICM Biometric Unit using a computer program generated with Stata software version 12 (StatCorp, LLC, College Station, TX, USA). Participants, interventionists, and assessors were not masked to group assignment. The control group was a usual care group without any diet or exercise intervention. No particular material was delivered to the control group during the intervention, and these patients were not asked to limit exercise practice or eat/avoid specific foods during the intervention period.

### 2.3. Intervention

The APAD education program, which was based on the previously published trial method [18], was implemented during chemotherapy and radiotherapy (26 weeks). The intervention included twice-weekly exercise sessions and six individual nutritional therapeutic education sessions. The exercise sessions were individually supervised hospital-based exercise sessions and non-supervised home-based sessions combining one muscle strength session and one aerobic session each week (Figure 1). The nutritional education sessions targeted body weight control and the modification of feeding behaviors according to the WCRF recommendations [25]. The nutritional sessions were planned on the same days as supervised hospital-based exercise sessions. The intervention was tailored to maintain the patients’ exercise level and dietary intake in accordance with the guidelines throughout the intervention period.

#### 2.3.1. Exercise

The exercise program delivered by trained professionals combined aerobic and muscle strengthening exercises according to international recommendations: 120 min of moderate to vigorous physical activity per week associated with strength training [9,26,27,28]. One muscle strength session and one aerobic session were scheduled each week (10 min of warm-up, at least 30 min of exercise, 10 min of stretching and 10 min of relaxation time with a goal of muscle recovery and well-being). Strength sessions targeted six main muscle groups (hamstrings, quadriceps, buttocks, abdominal, back, shoulders/arms), and each skill was performed for 2 to 5 sets with 6 to 12 repetitions with individual adaptation and progression. Aerobic exercise was performed at moderate intensity and adapted to the patient’s physical condition and progression, with a target of 50–75% of the maximum heart rate for 30 to 45 min (adapted to the patient’s physical condition). The initial exercise intensity was individualized but generally began at 50–55% of the maximum heart rate and progressed to 65–75% of the maximum heart rate by weeks 20 to 26. Supervised hospital-based sessions were achieved on a cycloergometer. For home-based practice, patients were given the option of various modalities of aerobic exercise (e.g., walking, jogging, cycling, dancing/fitness, swimming) to sustain adherence to the program and promote enjoyment. Hospital-based supervised exercise sessions aimed to provide the patients with relevant instructions that allow reproducibility at home and increased autonomy. Every supervised session was based on theory-based behavioral targets and techniques to improve behavioral change and patient adherence. Hospital-based supervised exercise sessions were scheduled on the same day as chemotherapy and during radiotherapy every 3 weeks to avoid additional cost. A total of eight hospital-based supervised exercise sessions and 44 at-home sessions were planned during the course of the intervention. Non-supervised, home-based sessions were planned at least twice per week, except only one home-based session was scheduled for the weeks that included one supervised hospital-based exercise session. Precise written instructions were given to patients in the educational and personable APAD-Moving workbook, which included information on their disease and reasons for being physically active during active treatment for cancer, illustrated instructions on performing the home exercises, the schedule for planned home-based sessions, and a patient log to evaluate adherence. Patients were asked to fill in the adherence log at home with whether planned sessions were achieved, the number of achieved muscular exercises, duration of each session, rating of perceived exertion (on scale of 1 to 10), reason for missed sessions, and anything else that they would like to discuss with the exercise specialist at the next supervised session.

#### 2.3.2. Diet

Patients in the intervention arm received diet counselling with therapeutic education from a dietician at six individual face-to-face sessions. Each session lasted approximately 30 min. During chemotherapy, four diet sessions were scheduled to achieve balanced dietary intake, advising patients on controlling weight and managing with the potential toxicities and side effects of chemotherapy. Two more sessions were planned at the beginning and end of radiotherapy for all intervention groups. Weight control was pursued in patients with BMI < 30 kg/m^2^; weight normalization was targeted in patients with BMI ≥ 30 kg/m^2^ (i.e., to decrease BMI to less than 30 kg/m^2^ by the end of adjuvant therapy). Each consultation involved an evaluation of nutritional status, nutrition care tailored to the patient’s caloric needs and potential toxicities related to treatment, and nutritional education. The purpose of these consultations was to teach the principles of a well-balanced diet, foster weight control during treatment, and induce appropriate feeding behaviors after treatment.

#### 2.3.3. Evaluation of Nutritional Status

At the first session, nutritional status was evaluated based on the patient’s usual weight, current weight, and their weight measured 1 to 6 months prior to study enrollment according to the French National Authority for Health criteria. The dietician assessed the patient’s daily energy requirement by computing their basal metabolic rate (BMR) [29] according to the corrected formula of Harris and Benedict. Dietary intake was prospectively measured by asking patients to fill out a questionnaire on 3 consecutive days of food intake at the first and last session. For the other sessions, dietary intake was measured by a 24-h recall food survey and a 10-point visual analogue scale.

#### 2.3.4. Nutrition Care

Nutrition care aimed for weight control through balanced dietary intake tailored to the patient’s energy needs and potential toxicities related to the cancer treatment. The dietician verified the patient’s intake utilizing the following guidelines: daily energy intake was compared to the estimated daily energy needs, patients were guided to regularly distribute their dietary intake into three main meals with an optional snack in the afternoon, macronutrient distribution was compared to the French dietary reference intakes for a balanced diet (i.e., 30–35% lipids, 50–55% carbohydrates, and 10–15% protein) [30], and food group intake was guided to meet the recommendations of the WCRF [25]. If the patient’s habits did not correspond with these guidelines, or their daily energy intake was higher or lower than 10% of the estimated daily energy needs, the dietician counselled the patient on modifications regarding foods, nutrients, meals, and calorie distribution. If the patient’s BMI was >30 kg/m^2^ by the end of chemotherapy, a new weight goal was set to decrease the patient’s BMI to the range of 25 to 30 kg/m^2^. A new range of daily energy needs was then estimated with a corresponding distribution according to food group balance and the WCRF guidelines [25]. Patients were given a printed example of food groups, servings, and distribution that they may eat on a typical day. In the following sessions, the dietician computed the patient’s intake again and adapted the advice to the evolution of the patient’s intake. Specific advice was given to patients on the management of potential toxicities and side effects of chemotherapy.

#### 2.3.5. Nutritional Education

Nutritional education aimed to teach the patients the principles of a well-balanced and healthy diet based on WCRF guidelines [25], inform them of industrial food packaging, and fight preconceived ideas by using practical applications and educational games. Nutritional education was tailored to the patients’ habits and means, precariousness level, and cultural and social environment.

Session 1: Presentation of detailed well-balanced menus.

Session 2: Identify the nature and specific roles of food groups—food balance education based on the food pyramid presentation.

Session 3: Food balance education based on the “APAD fridge game” consisting of the elaboration of three balanced meals with the food provided on the picture (food and dish choices from proposed menus to obtain balanced meals in special contexts, such as picnic, fast food, or restaurant).

Session 4: Teach to read labels and food packaging—food balance education through examples of complex mixed dishes (e.g., lasagna).

Session 5: Evidence-based information on the relationship between nutrition and cancer using a quiz game pointing out preconceived ideas.

Session 6: Post-treatment diet benefits and recommendations, and delivery of a booklet summarizing dietary WCRF recommendations.

#### 2.3.6. Missed Sessions

Missed supervised exercise or diet counseling sessions at the hospital during chemotherapy could not be rescheduled, as the patients only came to the hospital once every 3 weeks during chemotherapy. In the case of a missed supervised session, a phone call was made to the patient by the exercise and/or diet specialists. The discussion focused on reasons for not attending the session, patient adherence in the last 3 weeks, encouraging the patient to attend future exercise or diet counseling sessions, taking into account the difficulties, and delivering education targets and content of the missed session if possible.

In contrast, missed supervised hospital-based sessions during radiotherapy were rescheduled as soon as possible because most of the patients came to the hospital every weekday during radiation therapy.

### 2.4. Outcomes and Assessments

The endpoints and assessments were concordant with those of the previous APAD1 trial [18,20]. Endpoints included subjective PROs and objective outcomes. Assessments were conducted at each site at baseline, just before the start of adjuvant chemotherapy (T0); the end of chemotherapy (T1); end of radiotherapy (T2); and the 6-month follow-up (T3). The primary endpoint was self-reported cancer-related fatigue assessed by the General fatigue subscale of the Multidimensional Fatigue Inventory (MFI-20) [31,32], a 20-item self-report instrument that covers five dimensions. The other four subscales (Physical fatigue, Mental fatigue, Reduced activity, and Reduced motivation) and the total score were considered secondary outcomes.

An objective measure of physical fatigue, lower limb muscle endurance, was measured using the sit-to-stand test at 15 and 30 s. The adherence to chemotherapy was evaluated using the relative dose intensity (RDI), which was calculated as the ratio of the cumulative dose intensity (mg/m^2^/week) to the dose intensity planned in the chemotherapy protocol. In addition, anxiety and depression symptomatology was evaluated using the 14-item Hospital Anxiety Depression Scale (HADS) self-report questionnaire [33]. Quality of life (QoL) was assessed by the EORTC QLQ-C30 questionnaire, a validated cancer-specific instrument [34] evaluating five functions (physical, role, cognitive, emotional, and social), nine symptoms (fatigue, pain, nausea and vomiting, dyspnea, loss of appetite, insomnia, constipation, diarrhea, and financial difficulties), and global health status. Physical activity was assessed using the 16-item Global Physical Activity Questionnaire (GPAQ) developed by the World Health Organization (WHO) [35,36]. The GPAQ assesses the intensity, duration, and frequency of physical activity in a usual week in three domains: activity at work, travel to and from places, and recreational activities. Physical activity is then expressed in terms of metabolic equivalent (MET), which is the ratio between the speed of metabolism during physical activity and the speed of metabolism at rest [37]. MET values are applied to vigorous and moderate intensity variables in work and recreational settings. One MET is defined as 1 kcal/kg/h and is equivalent to the energy cost of sitting quietly. We attribute 4 MET and 8 MET to the time spent on moderately intense and vigorous physical activity, respectively. Outcomes considered were average METs (MET/min/day) from activities of moderate and vigorous intensity (work and recreational); average METs from moderate intensity transport (cycling and walking), total physical activity (MET-minutes/week), and sedentary time (min/day). We also calculated the proportion of patients with a low level of total physical activity and the proportion of patients that failed to meet the WHO recommendations. Anthropometric measures (body weight, height, BMI, and waist circumference) were used to describe weight gain. Dietary intake was evaluated using a food record [38] of the foods and beverages consumed for 3 consecutive days (including one weekend day); the data were entered into nutritional analysis software and calories and nutrient intake computed (Nutritional Analysis Software, release 8, Villiers-les-Nancy: MICRO 6, 2007). All endpoints, except the nutritional evaluation, were assessed at four time points: pre-intervention, baseline assessment before the start of adjuvant chemotherapy (T0); end of chemotherapy (T1); end of radiotherapy (T2; i.e., immediately post-intervention); and 6 months after the end of treatment (i.e., 1 year after inclusion in the study). The nutritional evaluation was conducted at three time points: T0, T2, and T3.

### 2.5. Statistical Considerations

#### 2.5.1. Sample Size Calculation and Randomization

The sample size calculation was based on the primary endpoint, the General fatigue subscale of the MFI-20 questionnaire. Smets et al. [31] estimated a mean score of 16 (standard deviation [SD] 8) for the General fatigue subscale in cancer patients undergoing radiotherapy; therefore, we considered this the reference value and the main time point of the evaluation was T2. In order to detect a 4-point reduction in the mean score on the General fatigue subscale in the intervention group with respect to the control group (i.e., reduction of 25% of the reference), the sample size calculation was based on a global risk alpha of 1% (bilateral situation), 90% (1-β) power, SD of 9, and the consideration of a repeated measures design (one pre-intervention measure and roughly three post-intervention measures, with a hypothetical correlation coefficient between measures of 0.2). The sample size was estimated to be 161 patients per arm in Stata Statistical Software, Release 10 (StataCorp, College Station, TX, USA). Considering a 12% loss to follow-up, 180 patients were required per group, a total of 360 patients overall in the trial. Randomization (1:1) was achieved using the minimization method and patients stratified according to two factors: the level of socioeconomic deprivation assessed by the French EPICES score [21,22] and recruiting center. Randomization was centralized at the Biometrics Units of the ICM, Montpellier.

#### 2.5.2. Statistical Analysis

All analyses were performed in the intention-to-treat population. Analyses related to the impact of the program were also performed in the per-protocol population. This population considers all eligible patients treated and evaluated, which in the case of the intervention arm was defined as all patients who completed at least one supervised or unsupervised exercise session. An initial descriptive analysis of the baseline variables was performed and balance between treatment groups checked for the main demographic, socio-professional, clinical, and PRO variables.

The efficacy of the program was evaluated by the relative difference in the General fatigue score at each time point (T1, T2, and T3) with respect to T0 and compared between two groups using the Kruskal–Wallis test. A model approach was also used; the evolution of the General fatigue score over time was assessed using a linear mixed model (LMM). Random intercepts and random slopes were included to take into account the time effect. Interaction terms were also considered. The model coefficients were estimated by maximum likelihood; coefficients are presented as β_1_ for arm effect (APAD with respect to control) and as β_0_ for time effect. Separate models were adjusted for each other of the MFI-20 questionnaire subscales and for all other secondary endpoints.

The EORTC QLQ-C30 was analyzed following the EORTC guidelines [39], with scores for each functional and symptom subscale. HADS scores assessing anxiety and depression were categorized according to Zigmond classification (absence of disorder, suspected disorder, disorder).

Physical activity data recorded with the GPAQ were analyzed according to the WHO guidelines and using the STEPS analysis syntax program for cleaning and analyses.

Categorical and ordinal variables were compared using the Pearson chi-squared or Fisher’s exact tests. Continuous variables were presented as mean and standard deviation (SD) and were compared using the non-parametric Kruskal–Wallis test.

In the case of missing values, no imputation method was used. All reported *p*-values are two-sided, and a significance threshold of 0.05 was considered. Statistical analyses were performed using the STATA 13 software (StataCorp LP, College Station, TX, USA).

### 2.6. Ethics Approval, Consent to Participate and Trial Registration

The study was approved by the French institutional review board (i.e., Comité de Protection des Personnes Sud-Méditerranée III; N°ID-RCB: 2012-A01648-35), the Agence Nationale de Sécurité du Médicament et des produits de santé (N°ANSM: 130313B-12), and the Commission nationale de l’informatique et des libertés. Written informed consent was obtained from all participants. The trial was registered under the identification number NCT04109326.

## 3. Results

From May 2013 to December 2014, 360 patients were randomized at eight centers to the APAD (*n* = 180) and control (*n* = 180) arms. PROs were collected from 99%, 85%, 81%, and 71% of the 360 randomized patients at T0, T1, T2, and T3, respectively (Figure 2). Thirty-eight patients in the APAD arm (21%) did not complete any of the supervised or unsupervised exercise sessions. Among the per-protocol APAD population (*n* = 142), 67 patients (47.2%) had 80% adherence to the exercise program (defined as the completion of at least 80% of supervised and 80% of unsupervised sessions); 93 patients (65.5%) had global 80% adherence (both modalities confounded).

The baseline characteristics of the patients are presented in Table 1. Most of the patients completed the study (80% in the APAD arm, 87.7% in the control arm). The study was discontinued early for 17% and 7.2% of the patients in the APAD and control arm, respectively, due to patient withdrawal from the study (*p* = 0.014).

### 3.1. Fatigue

Compared to T0, the median relative difference in General fatigue scores at T1 was 25% in the control arm and 21% in the APAD arm, which is not a significant difference between the two arms (*p* = 0.274; Table 2). At T2, the increase in fatigue was greater in the APAD arm (20%) than in the control arm (8%), but the difference was still not significant (*p* = 0.157). However, at T3, 1 year after inclusion, the increase in fatigue in the APAD arm (15%) was lower than that of the control arm (20%), though it was not statistically or clinically significant (*p* = 0.933, Figure 3). According to the adjusted model, general fatigue tends to increase over time (β_0_ = 0.024 [95% CI 0.01; 0.034]; *p* < 005) without an observable effect of the intervention in terms of a reduction in general fatigue over time (β_1_ = 0.33, *p* = 0.374; Table 2).

No significant benefit of the APAD program was observed when the analysis was conducted by precariousness level (EPICES score). However, we observed a clear gradient in the baseline fatigue according to the stratum of precariousness, with the fatigue level being higher for the most precarious stratum throughout the study (Appendix A).

For all other fatigue subs-scales of the MFI20, no difference was observed between treatment arms with respect to their evolution over time. Overall, physical and mental fatigue tended to increase across time in both arms (β_0_ = 0.02 and β_0_ = 0.03, *p* < 0.05; Table 2). Notably, a baseline imbalance disfavoring the APAD arm was observed for Mental fatigue (*p* = 0.019) and Reduced motivations (*p* = 0.025).

When the fatigue sub-scales were analyzed in the per-protocol population (Appendix A), we observed that the APAD program had a positive impact, enhancing motivation over time despite the initial imbalance (β2 = −0.023 [95% CI −0.39; −0.008]; *p* = 0.003), and reducing mental fatigue (β2 = −0.020 [−0.04; 0.002], *p* = 0.069).

In a sub-group analysis considering the 80% adherent population in the APAD and control arms, we found no significant difference in the relative differences in General fatigue scores (primary endpoint) (data not shown).

### 3.2. QoL and Psychological Distress

Quality of life and psychological distress scores on the EORTC QLQ-C30 and HADS, respectively, at baseline and during follow-up are described in Table 2 and Table 3. At the end of radiotherapy, we found no significant difference between arms in terms of QoL (Table 2, Appendix A). Overall, emotional function (both arms together) increased (LMM coefficient β_0_ = 0.004 [0.002; 0.005], *p*< 0.01) and cognitive function decreased (β_0_ = −0.002 [−0.004; −0.001], *p* < 0.01) over time. The symptom fatigue also increased (β_0_ = 0.004 [0.001; 0.007], *p* = 0.018), which is consistent with the results observed in the MFI score analysis. No impact of the APAD program was observed in regard to improving QoL dimensions global health status, functional dimensions, and fatigue symptoms.

When the analysis was conducted on the per-protocol population (Appendix A), we observed slightly better physical function at the end of radiotherapy (*p* = 0.043), but the LMM did not show a global trend over time for an impact of the APAD program on this function (β1 = 0.025 [−0.034; 0.084]; *p* = 0.800).

In a sub-group analysis considering the 80% adherent population in the APAD and control arms, we observed significantly better physical function in the APAD subgroup at T2 (*p* = 0.003) and T3 (*p* = 0.017). However, in that subgroup analysis, a baseline imbalance favored the 80% adherent APAD subgroup, which exhibited better physical function (*p* = 0.028) and lesser pain symptom (*p* = 0.002) at T0. This may be explained, in part, to better adherence to the program.

Regarding psychological distress (Table 3), a lower proportion of patients with confirmed depression (score >10) was observed in the APAD arm (54.03%) with respect to control (66.92%) at T3 (*p* = 0.052). A similar result was observed in a per-protocol analysis (*p* = 0.026; Appendix A).

### 3.3. Physical Activity

Dimensions of the GPAQ and sit-to-stand test are described in Table 4.

The intervention had a positive impact on the total MET (Figure 4) and on the moderate intensity recreational activities (Figure 5), as they were significantly higher in the APAD arm with respect to control (β_1_ = 0.74 [0.37; 1.10]; *p* < 0.001 and β_1_ = 0.96 [0.46; 1.43]; *p* < 0.001). However, the sitting or reclining time per day appeared to be slightly higher in the APAD arm (β_1_ = 0.18 [0.02; 0.33]; *p* = 0.023; Appendix A).

Regarding compliance with WHO recommendations (Table 5), patients in the APAD arm more frequently met the recommendations compared to the control arm at T1 (81.40% vs. 61.87%, *p* < 0.001) and T3 (86.60% vs. 68.14%, *p* = 0.002).

### 3.4. Dietary Intake

The evolution of dietary intake and weight control by randomization arm is summarized in Table 6 and Appendix A. At T2, a significant increase in the consumption of fiber was observed in the APAD arm (*p* = 0.020), as well as a reduction in the consumption of animal proteins and alcohol (*p* = 0.003 and *p* = 0.019, respectively). However, both reductions seem to have been only temporary, as 6 months later the effect was diminished. These results were confirmed when we analyzed the evolution of these parameters over time in an LMM.

According to the adjusted LMM, the intervention had a positive significant impact on increasing fiber consumption over time (β_1_ = 0.096 [0.026; 0.17]; *p* = 0.007). A trend of reduced consumption of animal proteins was also observed and was lower in the APAD arm than the control arm (β_1_ = −0.26 [−0.55; 0.017]; *p* = 0.066). No change in weight over time was observed.

### 3.5. Chemotherapy Completion Rates

The RDI was high overall, with 90.7% and 80% of the patients having an RDI > 80% and 90%, respectively, in the intention-to-treat population. No difference was found between the two arms (80.79% vs. 79.21%, *p* = 0.71 for RDI > 90%; 91.53% vs. 89.89%, *p* = 0.595 for RDI > 80% in APAD and control arms, respectively).

## 4. Discussion

In the present study, we reported the results of a large, multicenter, randomized trial evaluating the impact of an APAD education program on fatigue, fat mass, and health-related QoL. The results do not support our previous conclusions [18,20] and the main hypothesis of the efficacy of the exercise and diet education program to relieve cancer-related fatigue in EBC patients receiving adjuvant treatments. One of the main pitfalls could be linked to patient adherence to the protocol. The drop-out rate in the present study was higher in the APAD arm, with 17% of the population discontinuing the program early. In addition, the intensity of physical activity appears to have insufficient autonomy, with low adherence to the intervention. Twenty-one percent of the patients did not perform the complete supervised exercise sessions, and only 65% had completed at least 80% of the home sessions. Our intervention appears to change low activity only. Thus, this education program appears to be more of a lifestyle change, in terms of adherence and clinical impact, to more active practice, as confirmed by the high proportion of patients achieving the WHO targets without achieving a level of physical activity high enough to induce a clinically significant reduction in fatigue. The design of our program, with discontinuous supervision, may explain the ambiguous results for fatigue during follow-up [40,41,42,43,44,45].

As reported in recent meta-analyses and guidelines, supervised exercise interventions appear to have significantly greater effects on fatigue than unsupervised exercise interventions, and shorter supervised interventions with a duration ≤12 weeks appear to induce greater effects on fatigue than supervised interventions with a longer duration [9,46]. However, the results indicate significant benefits on depressive symptoms. This impact supports a reported decrease in psychological distress and an improvement in self-esteem with exercise [9].

In contrast to our hypothesis, we found no significant effect of the intervention on BMI and the chemotherapy completion rates. Previous reports on combined diet and exercise interventions delivered during chemotherapy [16,17] failed to find any benefits on hip circumference and BMI. One explanation could be based on the significant, but transient, modification of nutritional intake, specifically animal proteins, alcohol, and fiber, without persistent marked behavioral modifications, as highlighted by the trends in protein and lipid consumption in the LMM. Evaluation of the impact of continued education programs, with later-time supervised sessions, could help define the best method of sustaining early changes.

A meta-analysis of 32 randomized studies comprising 2626 EBC patients evaluated the impact of supervised aerobic or resistance exercise interventions during adjuvant treatment and reported a pooled significant improvement in strength [47]. High-dose training or a focus on resistance training appears to be associated with better effects on strength in this patient population [41,45,48]. At the same time, health education interventions appear to be associated with a lesser impact on cancer-related fatigue compared to exercise training [49].

Regarding behavioral outcomes after combined interventions including diet and exercise components delivered during chemotherapy, two studies [16,17] yielded significant changes in dietary intake and one presented significant changes in total physical activity in the intervention group (post- vs. pre-intervention) [17]. Our APAD intervention had a significantly favorable impact on leisure time physical activity at the end of chemotherapy and the end of radiotherapy, but improvements in total physical activity were not significant. Physical activity done in the framework of the APAD intervention was reported in the leisure category, which explains the enhancement of that physical activity type in our study. Regarding dietary intake, no significant changes were observed in the APAD arm of our study versus usual care. The 3-day record method and food to nutrition conversion software we used generated large standard deviations (see Table 4, baseline values for nutrients) that possibly impaired the statistical power to detect a between-group difference. The two previous studies [16,17] that demonstrated dietary changes analyzed dietary data that were collected by food frequency questionnaires.

Few studies evaluating a diet and/or exercise intervention have included follow-up measures [40,41,42,43,44]. Most of them, such as our present study, failed to show that significant effects were maintained after the end of the intervention [40,42,44,45]. One study reported improvements in the 6-min walk test 6 months post-intervention [43]. Difficulties could be related to the weak supervision and support over time, making patients unable to maintain the initially induced changes in behavior. The impact of the intervention seems to be limited in time, even for trials evaluating different doses of exercise in this context [50]. This finding may promote the necessity of setting longer supervised intervention models that can include, for example, the present “in-treatment” module during the chemotherapy part of the treatment plan, followed by additional supervised sessions during radiotherapy and a 6-month internet-based “survivor” module designed to maintain behavioral changes and support autonomy with limited cost. Several telephone- or internet-based diet–exercise interventions have been tested in BC survivors and yielded health benefits [51,52,53] with moderate to good adherence rates (from 41–87% of adherent patients) [54]. However, these adherence rates pinpoint the need for enduring strategies to maintain the motivation of this “survivor” population. In the study by Stone et al., adherence was significantly associated with moderate levels of exercise interventions (similar to the activity levels selected in our present study), BMI, physical health, and employment status, with reduced adherence in women working full-time compared to those who do not work full-time [55]. These findings, combined with the difference in general fatigue found in the present study, encompass the impact of social inequalities on the global impact of cancer and cancer treatments, as well as access and adherence to supportive care programs in these populations.

Our study was designed primarily to validate our previous results by assessing, in a pragmatic context, the effectiveness of an education program on exercise and diet compared to the standard of care in France (the usual care control arm). Though pragmatic and comprehensive, the main drawback of this design is that it does not allow disentanglement of the independent effects of exercise and dietary components. Another limitation is the differential drop-out rate in the APAD vs. usual care groups. Although many outcomes improved significantly in the APAD arm, and an increase was observed in the declared leisure physical activity, the between-group difference was not significant for objectively measured physical activity. The spontaneous physical activity level of the usual care group may have partly diluted the effects of the APAD intervention. Given the number of comparisons we made at each time point for the secondary outcomes without adjusting for multiple testing, we would expect a few false discoveries by chance.

Finally, based on the publication of Smets et al. [31], we hypothesized that our population would be affected by significant levels of cancer- and treatment-induced fatigue. However, our population was affected by much lower levels of fatigue than the expected mean Global fatigue score of 16, as the mean General fatigue scores were 9.95 and 9.63 for the APAD and control arms, respectively. These relatively low levels of fatigue could explain some of the negative results. Buffart et al. recently reported in a meta-analysis that, even if exercise induced a benefit for all patients during treatment, only patients affected by the worse levels of fatigue experienced a persistent impact of exercise post-treatment [56]. Thus, targeting specific subgroups of patients with higher scores for fatigue or variables associated with a greater level of fatigue during treatment may be more beneficial and cost-effective. In this regard, the observation of a gradient in the level of fatigue throughout the treatment period according to the stratum of precariousness confirms that the level of precariousness evaluated using the EPICES score is a stratification factor and fully justifies its inclusion as a stratification factor in this study. It seems important to consider this social parameter in the design of future studies evaluating the impact of supportive care interventions during the treatment of EBC.

## 5. Conclusions

In French patients receiving adjuvant treatment for EBC, a mixed hospital- and home-based diet and exercise education program during adjuvant chemotherapy and radiation therapy inducing a transient modification in the level of physical activity and dietary intake is not sufficient to demonstrate improvements in fatigue during and after treatment. We found no impact on the dose intensity of chemotherapy. Fatigue appears to correlate with the level of precariousness. Cancer care centers should consider integrating more proactive diet–exercise supportive care into the management of patients with BC who are receiving chemotherapy and/or radiotherapy, focusing on fatigued and/or precarious patients. Additional information is needed in order to identify the optimal intervention timing to induce persistent lifestyle changes and reduce long term alterations of patients’ quality of life in this setting.

## Figures and Tables

**Figure 1 nutrients-12-03081-f001:**
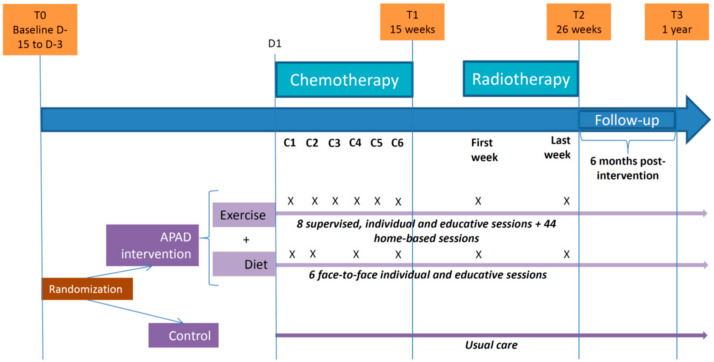
Flow diagram of the Adapted Physical Activity Diet 2 (APAD2) trial. C indicates chemotherapy cycle time points. X indicates intervention times in the associated randomization arm. Number of participating patients: T0 (Baseline): Control (*n* = 180), Adapted Physical Activity Diet (APAD) (*n* = 180). T2 (completed Chemotherapy and Radiotherapy sessions): Control (*n* = 170), APAD (*n* = 160). T3 (1 year after inclusion): Control (*n* = 157), APAD (*n* = 144).

**Figure 2 nutrients-12-03081-f002:**
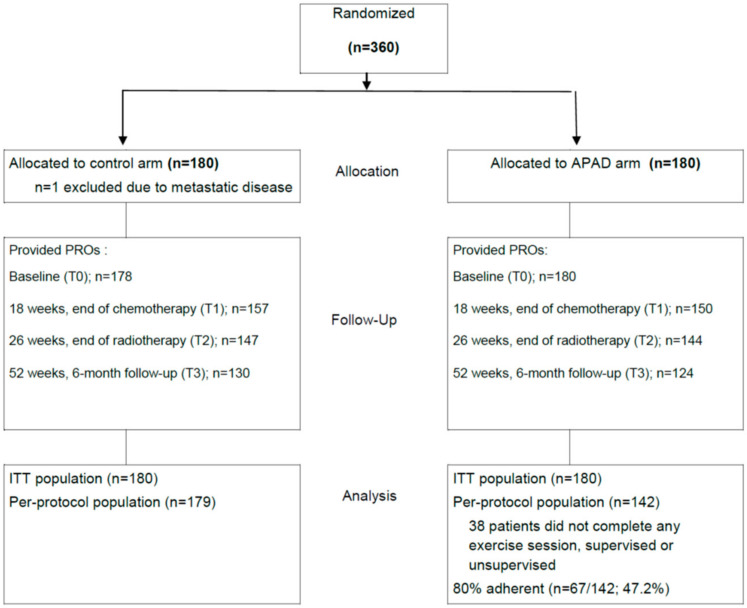
CONSORT diagram of the study.

**Figure 3 nutrients-12-03081-f003:**
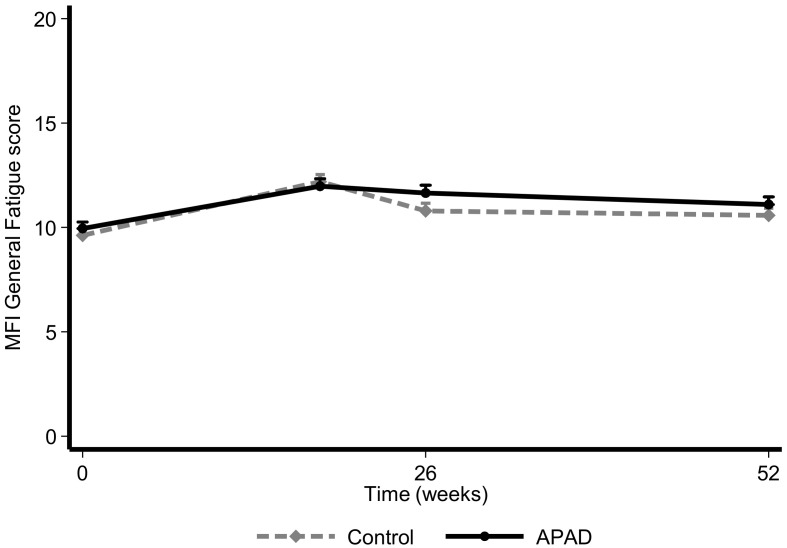
Evolution of the MFI General Fatigue score according to randomization arm in the intention-to-treat population. Data are presented as mean + SD.

**Figure 4 nutrients-12-03081-f004:**
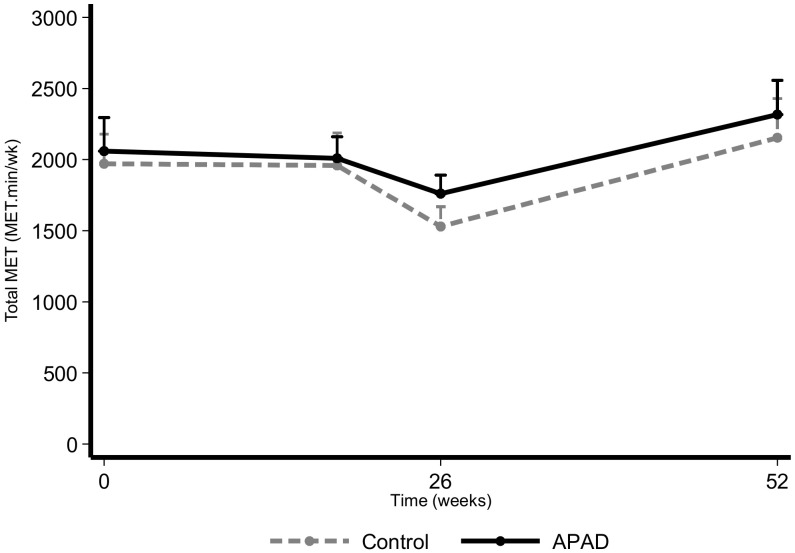
Evolution of the total MET based on the GPAQ according to randomization arm in the intention-to-treat population. Data are presented as mean + SD.

**Figure 5 nutrients-12-03081-f005:**
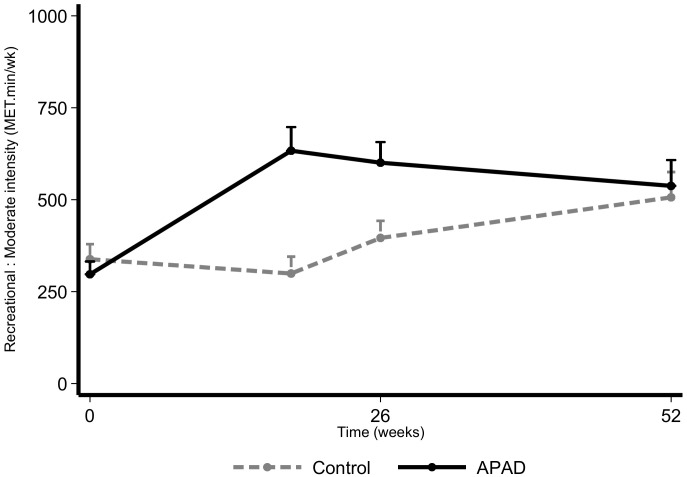
Moderate intensity recreational MET over time based on the GPAQ according to randomization arm in the intention-to-treat population. Data are presented as mean + SD.

**Table 1 nutrients-12-03081-t001:** Baseline characteristics of patients in the intention-to-treat population.

	Control *n* = 180		APAD *n* = 180		Total *n* = 360	
	Mean	SD	Mean	SD	Mean	SD
Age (years)	52.35	10.09	52.66	9.69	52.51	9.88
Weight (kg)	67.00	14.13	68.41	14.60	67.71	14.36
BMI (kg/m^2^)	25.22	5.30	25.72	5.14	25.47	5.22
BMI categories	*n*	%	*n*	%	*n*	%
<18.5 kg/m^2^	6	3.33	2	1.12	8	2.23
18.5–24.9 kg/m^2^	99	55.00	95	53.07	194	54.04
25–29.9 kg/m^2^	45	25.00	45	25.14	90	25.07
≥30 kg/m^2^	30	16.67	37	20.67	67	18.66
Post-menopausal	88	48.89	88	48.89	176	48.89
Tobacco smoking						
Non-smoker	102	56.67	86	47.78	188	52.22
Smoker	29	16.11	34	18.89	63	17.50
Ex-smoker	49	27.22	60	33.33	109	30.28
Marital status						
Single/divorced/widowed, no child	16	8.99	10	5.56	26	7.26
Single/divorced/widowed, with child	23	12.92	37	20.56	60	16.76
Married/living together, no child	21	11.80	26	14.44	47	13.13
Married/living together, with child	118	66.29	107	59.44	225	62.85
Education level						
No qualifications	29	16.57	24	13.56	53	15.06
Secondary level	43	24.57	31	17.51	74	21.02
Completed high school	29	16.57	43	24.29	72	20.45
Completed ≥ 2 years at university	74	42.29	79	44.64	153	43.47
Usual professional status						
Full or part-time employed	97	53.89	103	57.22	200	55.56
Retired	42	23.33	41	22.78	83	23.06
Unemployed/medical leave	41	22.78	36	20.00	77	21.38
EPICES precariousness (or deprivation) level						
Non-precarious	109	60.56	109	60.56	218	60.56
Intermediate	60	33.33	60	33.33	120	33.33
Precarious	11	6.11	11	6.11	22	6.11
Surgery type	*n*	%	*n*	%	*n*	%
Lumpectomy	89	49.44	88	48.89	177	49.17
Quadrantectomy	37	20.56	45	25.00	82	22.78
Mastectomy	54	30.00	46	25.56	100	27.78
T stage						
T1	91	50.56	97	53.89	188	52.22
T2	74	41.11	73	40.56	147	40.83
T3	11	6.11	8	4.44	19	5.28
T3	3	1.67	1	0.56	4	1.11
T4	1	0.56	0	0	1	0.28
Tis	0	0	1	0.56	1	0.28
T stage						
N0	71	39.66	79	44.63	150	42.13
N1	86	48.04	83	46.89	169	47.47
N2	14	7.82	11	6.21	25	7.02
N3	7	3.91	3	1.69	10	2.81
NX	1	0.56	1	0.56	2	0.56
Breast cancer subtype						
Triple negative	17	18.48	17	18.89	34	18.68
HER2+, ER+, and/or PR+	29	31.52	35	38.89	64	35.16
HER2+, ER−, and PR−	9	9.78	10	11.11	19	10.44
HER2−, ER+, and/or PR+	37	40.22	28	31.11	65	35.71

**Table 2 nutrients-12-03081-t002:** Fatigue sub-scales of the MFI20 and quality of life (EORTC QLQ-C30) in the intention-to-treat population.

		Baseline (T0)		End of CT (T1)			End of RT (T2)			1 Year after Inclusion (T3)			LMM Coefficients ^1^ (95% CI)
		Mean	SD	Mean	SD	*p*	Mean	SD	*p*	Mean	SD	*p*	
General fatigue (Endpoint)	Control	9.63	4.2	12.19	4.21	0.683	10.79	4.44	0.107	10.58	3.84	0.231	β_1_ = 0.33 [−0.40; 1.07], *p* = 0.374
	APAD	9.95	4.18	11.97	4.37		11.65	4.44		11.1	4.07		β_0_ = 0.24 [0.014; 0.034], *p* < 0.05
*Median (range) of the relative difference* ^2^	Control			0.25 (−0.67; 3.0)			0.083 (−0.60; 3.50)			0.20 (−0.69; 3.5)			
	APAD			0.21 (−0.5; 2.75)			0.20 (−0.58; 3.25)			0.15 (−0.6; 3.0)			
				*p* = 0.274			*p* = 0.157			*p* = 0.933			
Physical fatigue	Control	9.55	3.72	11.28	4.05	0.896	10.61	3.96	0.985	10.1	3.63	0.742	β_1_ = −0.15 [−0.81; 0.52], *p* = 0.670
	APAD	9.11	3.85	11.33	4.19		10.64	4.25		10.06	3.77		β_0_ = 0.02 [0.007; 0.025], *p* < 0.05
Mental fatigue ^3^	Control	7.66	3.81	9.19	4.26	1	8.9	4.31	1	8.82	4.26	0.872	β_2_ = −0.015 [−0.035; 0.005], *p* = 0.152
	APAD	8.57	4.07	9.15	4.24		9.06	4.71		9.06	4.65		β_0_ = 0.03 [0.013; 0.042], *p* < 0.05
Reduced activities	Control	8.34	3.64	10.12	4.26	0.466	9.14	3.92	0.433	8.75	4.02	0.743	β_1 =_ 0.30 [−0.40; 0.99], *p* = 0.399
	APAD	9.01	4.01	9.84	4.64		9.67	4.49		8.9	4		β_0_ = 0.004 [−0.005; 0.014], *p* = 0.402
Reduced motivation ^3^	Control	7.73	3.5	8.22	3.92	0.587	7.84	3.46	0.651	8.07	3.19	0.572	β_1_ = −0.30 [−0.34; 0.93], *p* = 0.360
	APAD	8.54	3.73	8.31	3.7		8.15	3.77		8.1	3.83		β_0_ = 0.002 [−0.006; 0.009], *p* = 0.696
**EORTC QLQ-C30**													
Global health status	Control	69.94	18.67	59.39	21.22	0.516	66.55	19.53	0.429	67.44	19.3	0.537	β_1_ = −0.0026 [−0.071; 0.065], *p* = 0.940
	APAD	68.45	19.55	60.83	21.34		64.82	19.1		69.17	17.76		β_0_ = 0.001 [−0.001; 0.001], *p* = 0.824
Physical functioning	Control	87.17	14.59	79.79	19.1	0.219	84.4	15.38	0.083	85.45	17.26	0.194	β_1_ = −0.020 [−0.034; 0.075], *p* = 0.466
	APAD	88.48	14.64	82	19.58		86.49	17.11		89.46	12.95		β_0_ = −0.0001 [−0.0007; 0.0006], *p* = 0.873
Role functioning	Control	84.46	22.16	77.99	23.34	0.257	84.35	21.03	0.966	87.05	18.48	0.309	β_1_ = 0.026 [−0.073; 0.125], *p* = 0.607
	APAD	86.57	20.4	80.63	23.25		83.8	22.01		90.05	16.86		β_0_ = 0.0009 [−0.0005; 0.0022], *p* = 0.220
Emotional functioning	Control	63.7	23.52	72.75	24.83	0.211	75.51	23.09	0.309	73.26	20.1	0.679	β_1_ = 0.007 [−0.106; 0.119], *p* = 0.910
	APAD	63.66	23.95	70.5	22.93		73.1	23.28		73.21	23.33		β_0_ = 0.004 [0.002; 0.005], *p* < 0.01
Cognitive functioning	Control	85.21	20.5	79.3	25.14	0.657	80.16	21.75	0.849	79.97	23.42	0.957	β_1_ = −0.019 [−0.114; 0.076], *p* = 0.695
	APAD	84.17	20.32	78.56	24.37		79.58	22.4		80.24	23.12		β_0_ = −0.002 [−0.004; −0.001], *p* < 0.01
Social functioning	Control	82.58	24.15	66.77	30.63	0.827	73.58	27.71	0.23	82.04	23.99	0.849	β_1_ = 0.003 [−0.128; 0.134], *p* = 0.964
	APAD	84.45	21.97	66.44	29.73		69.84	28.42		83.6	21.78		β_0_ = −0.00008 [−0.002; 0.002], *p* = 0.942
Fatigue symptom	Control	28.54	21.86	44.87	27.24	0.934	34.62	25.73	0.152	32.26	21.96	0.935	β_1_ = −0.042 [−0.248; 0.164], *p* = 0.687
	APAD	28.49	23.35	45.38	29.13		37.4	23.8		31.63	21.15		β_0_ = 0.004 [0.001; 0.007], *p* = 0.018

^1^ In the linear mixed model (LMM; log transformed variables): β_1_ is the coefficient of the variable ‘arm’ (interpreted as APAD effect with respect to Control) noted as β_2_ when interaction (arm by time) was significant. β_0_ is the coefficient of the variable time (weeks) variable. ^2^ For each patient, the relative difference (RD) with respect to the baseline value of the General fatigue subscale at the end of chemotherapy (CT), end of radiotherapy (RT; end of the oncological treatment), and 1 year after inclusion was calculated as: (GFS__endRT −_ GFS__Inclusion_)/GFS__Inclusion_. A smaller RD indicates a greater reduction in general fatigue. ^3^ A baseline imbalance between arms was observed for Mental fatigue (*p* = 0.019) and Reduced motivations (*p* = 0.025). No other baseline imbalance was observed.

**Table 3 nutrients-12-03081-t003:** Anxiety and depression disorders in the intention-to-treat population.

	Control		APAD		
	*n*	%	*n*	%	*p*
**Baseline (T0)**					
Anxiety					0.662
Absence (<7)	0	0.00	0	0.00	
Suspected (8–10)	2	1.12	3	1.67	
Confirmed (>10)	176	98.88	177	98.33	
Mean anxiety (SD)	11.85 (2.56)		11.92(2.69)		
Depression					0.368
Absence (<7)	0	0.00	2	1.11	
Suspected (8–10)	70	39.33	71	39.44	
Confirmed (>10)	108	60.67	107	59.44	
Mean depression (SD)	18.93 (3.34)		18.78 (3.54)		
**End of chemotherapy (T1)**					
Anxiety					0.974
Absence (<7)	0	0.00	0	0.00	
Suspected (8–10)	1	0.64	1	0.67	
Confirmed (>10)	156	99.36	149	99.33	
Mean anxiety (SD)	12.29 (3.21)		11.99 (3.03)		
Depression					0.163
Absence (<7)	0	0.00	3	2.00	
Suspected (8–10)	55	35.03	57	38.00	
Confirmed (>10)	102	64.97	90	60.00	
Mean depression (SD)	20.25 (3.26)		20.46 (3.22)		
**End of radiotherapy (T2)**					
Anxiety					
Absence (<7)	0	0.00	0	0.00	
Suspected (8–10)	0	0.00	0	0.00	
Confirmed (>10)	147	100.00	142	100.00	
Mean anxiety (SD)	11.69 (3.04)		11.97 (3.04)		
Depression					0.576
Absence (<7)	1	0.68	1	0.70	
Suspected (8–10)	68	46.26	57	40.14	
Confirmed (>10)	78	53.06	84	59.15	
Mean depression (SD)	20.38 (3.06)		20.08 (3.34)		
**1 year after inclusion (T3)**					
Anxiety					0.367
Absence (<7)	0	0.00	1	0.81	
Suspected (8–10)	1	0.77	0	0.00	
Confirmed (>10)	129	99.23	123	99.19	
Mean anxiety (SD)	11.85 (2.42)		11.81 (3.11)		
Depression					0.052
Absence (<7)	0	0.00	2	1.61	
Suspected (8–10)	43	33.08	55	44.35	
Confirmed (>10)	87	66.92	67	54.03	
Mean depression (SD)	20.26 (2.87)		19.87 (3.62)		

**Table 4 nutrients-12-03081-t004:** Physical activity according to the GPAQ and muscular test in the intention-to-treat population.

		Baseline (T0)		End of CT (T1)			End of RT (T2)			1 Year after Inclusion (T3) (*n* = 113; *n* = 97)			LMM Coefficients ^1^ (95% CI)
		(158/168)		(*n* = 139; *n* = 129)			(*n* = 134; *n* = 128)						
		Mean	SD	Mean	SD	*p*	Mean	SD	*p*	Mean	SD	*p*	
Total MET (MET.min/wk)	Control	1998.63	2632.06	2339.37	4181.16	0.03	1522.69	1591.27	0.048	2157.1	2902.02	0.03	β_1_ = 0.74 [0.37; 1.10], *p* < 0.001
	APAD	2133.07	3206.8	2023.1	1729.05		1760.78	1464.05		2363.96	2386.1		β_0_ = −0.005 [−0.007; 0.007], *p* = 0.864
Recreational–Moderate intensity (MET.min/wk)	Control	338	511.97	299.08	539.7	<0.001	395.82	538.71	0.001	506.27	728.95	0.349	β_1_ = 0.95 [0.46; 1.43], *p* < 0.001
	APAD	296.98	454.64	632.87	730.3		600.56	630.25		537.2	694.62		β_0_ = 0.021 [0.012; 0.030], *p* < 0.001
Recreational–Vigorous intensity (MET.min/wk)	Control	122.78	461.56	219.86	829.16	0.433	173.73	618.49	0.636	216.28	777.94	0.353	β_1_ = 0.27 [−0.13; 0.67], *p* = 0.184
	APAD	135	450.99	187.6	538.19		220.31	767.45		274.06	654.44		β_0_ = 0.008 [0.001; 0.015], *p* = 0.019
Work—Moderate intensity (MET.min/wk)	Control	816.46	1479.06	515.11	1419.7	0.006	429.28	849.11	0.674	667.65	1423.57	0.743	β_1_ = 0.34 [−0.19; 0.86], *p* = 0.212
	APAD	824.4	1638.35	576.59	1001.46		508.78	998.82		546.1	963.79		β_0_ = −0.012 [0.022; −0.002], *p* = 0.014
Work—Vigorous intensity (MET.min/wk)	Control	212.41	1378.38	523.17	2778.69	0.935	53.73	316.49	0.813	120.35	804.99	0.721	β_1_ = 0.10 [−0.16; 0.36], *p* = 0.454
	APAD	374.29	1906.61	78.76	420.47		79.38	556.03		225.15	1119.72		β_0_ = 0.00003 [−0.005; 0.005], *p* = 0.989
Travel—Moderate intensity (MET.min/wk)	Control	508.99	816.59	782.16	1117.32	0.071	470.12	702.81	0.277	646.55	1307.34	0.141	β_1_ = 0.19 [−0.35; 0.72], *p* = 0.494
	APAD	502.4	805.68	547.29	964.76		465.23	542.61		781.44	1734.93		β_0_ = 0.01 [0.01; 0.02], *p* = 0.022
Sitting or reclining time (min/day)	Control	372.41	175.26	355.29	197.46	0.041	357.07	171.3	0.84	337.12	164.13	0.946	β_1_ = 0.18 [0.02; 0.33], *p* = 0.023
	APAD	400.85	178.62	391.05	177.33		369.2	177.26		353.76	183.5		β_0_ = −0.004 [−0.008; −0.0004], *p* = 0.029
**Muscular test**													
Sit-to-stand 30 s	Control	18.09	4.67	17.3	5.51	0.013	18.86	5.32	0.39	19.42	6.02	0.615	β_1_ = 0.001 [−0.002; 0.002], *p* = 0.094
													β_0_ = 0.001 [0.0002; 0.002], *p* = 0.017
	APAD	17.72	4.87	18.61	4.77		19.68	5.64		20.04	6.16		
Sit-to-stand ratio	Control	1.97	0.38	1.93	0.2	0.341	1.94	0.16	0.59	1.93	0.16	0.844	β_1_ = −0.002 [−0.014; 0.01], *p* = 0.694
(30 s/15 s)													β_0_ = −0.0001 [−0.0004; 0.0002], *p* = 0.431
	APAD	1.93	0.18	1.94	0.17		1.95	0.13		1.93	0.15		

^1^ In the linear mixed model (LMM; log transformed variables): β_1_ is the coefficient of the variable ‘arm’ (interpreted as APAD effect with respect to Control) noted as β_2_ when interaction arm by time was significant. β_0_ is the coefficient of the variable time (weeks) variable. Note: No baseline imbalance was observed for the studied variables (for sitting or reclining time (GPAQ), *p* = 0.136). CT: chemotherapy, RT: radiotherapy.

**Table 5 nutrients-12-03081-t005:** Compliance with the WHO Stepwise ^1^ recommendations for physical activity (GPAQ) in the intention-to-treat population.

		Control		APAD		Total		
		*n*	%	*n*	%	*n*	%	*p*
Baseline (T0)								
Low activity								0.084
	No	105	66.46	96	57.14	201	61.66	
	Yes	53	33.54	72	42.86	125	38.34	
Failed to meet WHO recommendations								0.568
	No	118	74.68	130	77.38	248	76.07	
	Yes	40	25.32	38	22.62	78	23.93	
End of chemotherapy (T1)								
Low activity								0.004
	No	71	51.08	88	68.22	159	59.33	
	Yes	68	48.92	41	31.78	109	40.67	
Failed to meet WHO recommendations								0.000
	No	86	61.87	105	81.40	191	71.27	
	Yes	53	38.13	24	18.60	77	28.73	
End of radiotherapy (T2)								
Low activity								0.094
	No	81	60.45	90	70.31	171	65.27	
	Yes	53	39.55	38	29.69	91	34.73	
Failed to meet WHO recommendations								0.071
	No	95	70.90	103	80.47	198	75.57	
	Yes	39	29.10	25	19.53	64	24.43	
1 year after inclusion (T3)								
Low activity								0.003
	No	64	56.64	74	76.29	138	65.71	
	Yes	49	43.36	23	23.71	72	34.29	
Failed to meet WHO recommendations								0.002
	No	77	68.14	84	86.60	161	76.67	
	Yes	36	31.86	13	13.40	49	23.33	

^1^ STEPS analysis program.

**Table 6 nutrients-12-03081-t006:** Dietary intake per day (3-day record) and weight control variables in the intention-to-treat population.

		Baseline (T0)			End of Radiotherapy (T2)			1 Year after Inclusion (T3)			LMM Coefficients ^1^ [95% CI]
		Mean	SD	*p*	Mean	SD	*p*	Mean	SD	*p*	
Total energy (Kcal)	Control	1507.46	362.87	0.312	1402.41	386.27	0.637	1382.97	384.02	0.16	β_1_ = −0.0020 [−0.054; 0.050], *p* = 0.947
	APAD	1463.76	434.09		1377.56	373.81		1452.03	372.93		β_0_ = −0.0010 [−0.002; −0.0002], *p* = 0.013
Animal proteins (g)	Control	19.64	17.92	0.233	10.38	10.04	0.003	8.35	8.99	0.233	β_1_ = −0.26 [−0.55; 0.017], *p* = 0.066
	APAD	17.29	17.03		7.95	9.94		7.42	8.55		β_0_ = −0.017 [−0.022; −0.011], *p* < 0.001
Vegetal proteins (g)	Control	8.85	5.93	1	5.61	4.29	0.415	5.49	4.3	0.836	β_1_ = −0.084 [−0.23; 0.067], *p* = 0.276
	APAD	9.2	6.87		5.45	5.59		5.53	5.07		β_0_ = −0.010 [−0.013; −0.007], *p* < 0.01
Lipids (g)	Control	61.43	21.98	0.147	57	22.9	0.22	54.82	24.05	0.407	β_1_ = −0.037 [−0.11; 0.034], *p* = 0.306
	APAD	57.72	23.49		52.32	18.73		56.34	19.54		β_0_ = −0.001 [−0.003; −0.0002], *p* = 0.021
Monounsaturated lipids (g)	Control	22	9.64	0.207	19.39	9.95	0.805	19.73	12.88	0.538	β_1_ = −0.024 [−0.11; 0.057], *p* = 0.558
	APAD	20.55	9.27		18.37	7.15		19.62	7.51		β_0_ = −0.002 [−0.003; 0.0004], *p* = 0.009
Polyunsaturated lipids (g)	Control	7.86	5.35	0.549	7.92	5.32	0.791	7.26	4.51	0.085	β_1_ = 0.022 [−0.078; 0.12], *p* = 0.671
	APAD	7.4	4.31		7.35	3.97		8.11	4.42		β_0_ = 0.0002 [−0.001; 0.002], *p* = 0.773
Simple sugars (g)	Control	69.97	28.08	0.926	61.93	23.26	0.273	61.26	24.46	0.504	β_1_ = 0.023 [−0.058; 0.10], *p* = 0.582
	APAD	70.25	30.15		65.66	24.96		64.03	26.17		β_0_ = −0.002 [−0.004; 0.001], *p* < 0.001
Alcohol (g)	Control	4.18	6.58	0.742	4.11	5.95	0.019	4.15	6.83	0.055	β_1_ = −0.091 [−0.010; −0.0009], *p* = 0.549
	APAD	4.44	7.31		2.25	4.18		2.98	5.67		β_0_ = −0.005 [−0.010; -0.0009], *p* = 0.020
Fiber (g)	Control	15.54	5.56	0.277	15.35	5.73	0.02	15.35	6.03	0.003	β_1_ = 0.096 [0.026; 0.17], *p* = 0.007
	APAD	16.28	5.87		17.37	6.49		17.58	5.9		β_0_ = 0.0005 [−0.0006; 0.001], *p* = 0.389
**Weight control**											
Weight (kg)	Control	67	14.13	0.44	66.23	12.93	0.571	67.13	13.98	0.576	β_1_ = 0.020 [−0.020; 0.060], *p* = 0.334
	APAD	68.41	14.6		67.57	13.56		68.28	14.19		β_0_ = 0.0001 [−0.0001; 0.0002], *p* = 0.389
BMI (kg/m^2^)	Control	25.22	5.3	0.317	24.95	5.01	0.51	25.23	5.28	0.655	β_1_ = 0.020 [−0.019; 0.059], *p* = 0.320
	APAD	25.72	5.14		25.29	4.79		25.47	5.01		β_0_ = 0.0001 [−0.0001; 0.0002], *p* = 0.337
Waist size (cm)	Control	163.08	6.49	0.717	87.09	13.05	0.827	87.26	15.74	0.344	β_1_ = 0.0034 [−0.027; 0.034], *p* = 0.827
	APAD	163.06	6.32		86.87	11.72		88.65	13.48		β_0_ = 0.0002 [−0.0001; 0.0004], *p* = 0.133

^1^ In the linear mixed model (LMM; log transformed variables): β_1_ is the coefficient of the variable ‘arm’ (interpreted as APAD effect with respect to Control) noted as β_2_ when interaction arm by time was significant. β_0_ is the coefficient of the variable time (weeks) variable. Note: No baseline imbalance was observed for the studied variables.

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
