# Peer review of "Brief Hospital Supervision of Exercise and Diet During Adjuvant Breast Cancer Therapy Is Not Enough to Relieve Fatigue: A Multicenter Randomized Controlled Trial"

_nutrients, 2020, doi:10.3390/nu12103081_

Round 1
Reviewer 1 Report
Thank you for letting me inspect this manuscript.
This is a well-performed RCT dealing with an important topic. RCT of this size are rare regarding nutritional interventions. The manuscript is well-organized and well-written, statistical tests are appropriate.
I want to raise the following points to clarify some statements regarding the intervention program:
- The detailed description of the primary outcome might be presented more clearly or more easily accessible. I needed to read the somewhat broad goal in the last para of the introduction, then go to lines 234ff to get more information, and finally consult the sample size section line 277 to understand that most probably a positive outcome would have been if there had been a 4 point difference in General fatigue subscale. Still, I am not quite sure about this. Thus, this would be nice to be presented with more transparency.
- Since many different programs have been suggested and are being studied on this topic, it should be made clear how the exercise/nutrition program was performed. I feel somewhat irritated by diverging statements on the number and duration of exercise sessions:
Line 123: “one aerobic session each week”
Line 135: “two aerobic sessions ... each week”
Line 133: “120 min of … activity per week”
Lines 135-136: “10 min warm-up, 30 min of exercise, 10 min of cool-down “
Lines 138-140: “Aerobic exercise was performed … for 30 to 45 minutes”
Differences in numbers and duration should be explained.
Author Response
Thank you for letting me inspect this manuscript.
This is a well-performed RCT dealing with an important topic. RCT of this size are rare regarding nutritional interventions. The manuscript is well-organized and well-written, statistical tests are appropriate.
I want to raise the following points to clarify some statements regarding the intervention program:
- The detailed description of the primary outcome might be presented more clearly or more easily accessible. I needed to read the somewhat broad goal in the last para of the introduction, then go to lines 234ff to get more information, and finally consult the sample size section line 277 to understand that most probably a positive outcome would have been if there had been a 4 point difference in General fatigue subscale. Still, I am not quite sure about this. Thus, this would be nice to be presented with more transparency.
Thank you for this comment. You are right. As the methods section is indeed an important one in this manuscript, we highlighted in the revised version the main objective, at the end of the introduction (L84) and at the beginning of the Materials and Methods section, in the design subsection (L92-99) to highlight the primary objective of the trial.
- Since many different programs have been suggested and are being studied on this topic, it should be made clear how the exercise/nutrition program was performed. I feel somewhat irritated by diverging statements on the number and duration of exercise sessions:
Line 123: “one aerobic session each week” and Line 135: “two aerobic sessions ... each week”
Line 133: “120 min of … activity per week”, Lines 135-136: “10 min warm-up, 30 min of exercise, 10 min of cool-down “ and Lines 138-140: “Aerobic exercise was performed … for 30 to 45 minutes”
Differences in numbers and duration should be explained.
Please accept our apologies for these discrepancies. Indeed, it was an error.
We corrected the first discrepancy by correcting to one aerobic session each week on L135. The exercise program was homogeneous, 2 sessions a week (one muscle strength session and one aerobic session).
Line 133: “120 min of … activity per week”, Lines 135-136: “10 min warm-up, 30 min of exercise minimum, 10 min of stretching and 10 min of relaxation time with the goal of muscle recovery and well-being “ and Lines 138-140: “Aerobic exercise was performed … for 30 to 45 minutes adapted to the patient’s physical condition”
Reviewer 2 Report
I would like to congratulate to the authors for this randomized study. Although results were not significant, it must be very hard to perform this kind of randomized study. As a physician who is treating breast cancer patients everyday, I would like to have authors consider to apply APAD after the completion of their treatment. Usually patients do not experience depression or fatigue much during the therapy. Most of patients realize or complain this emotional and physical burden after the completion of therapy.
Author Response
I would like to congratulate to the authors for this randomized study. Although results were not significant, it must be very hard to perform this kind of randomized study. As a physician who is treating breast cancer patients everyday, I would like to have authors consider to apply APAD after the completion of their treatment. Usually patients do not experience depression or fatigue much during the therapy. Most of patients realize or complain this emotional and physical burden after the completion of therapy.
Thank you for your interest in our study, and for your proposal. The question of the optimal timing for this information remains important and unresolved. We added in the conclusion the concept of intervention during treatment, and a sentence regarding the need for additional trials on the better moment to introduce this type of intervention during the adjuvant treatment. However, considering the primary objective of this study (general fatigue at the end of radiation therapy), an early intervention program was necessary.